# Effects of Rearing Management Applied throughout the Charolais Young Bulls’ Life on Carcass and Meat Quality

**DOI:** 10.3390/foods11182878

**Published:** 2022-09-16

**Authors:** Julien Soulat, Valérie Monteils, Brigitte Picard

**Affiliations:** Université Clermont Auvergne, INRAE, VetAgro Sup, UMR Herbivores, F-63122 Saint-Genès-Champanelle, France

**Keywords:** sensory qualities, rearing factors, rearing surveys, whole life

## Abstract

The aim of this work was to study, for the first time, the effects of the rearing management (from birth to slaughter) applied throughout the life of young bulls on carcass and meat quality. Five rearing managements were defined statistically, from a combination of 30 rearing factors, using a hierarchical clustering on principal components. This study considered the individual data of 179 Charolais young bulls from commercial farms. The carcass traits were more sensitive to rearing management than the meat traits. Rearing management had an effect mainly on fat and overall meat grain for the carcass, and on color and tenderness for the longissimus meat. However, it was possible to produce carcass and/or meat with similar properties from different rearing managements. Among the five rearing managements defined in this study, two were identified as allowing the best trade-off to produce simultaneously high carcass and meat quality. The first management was characterized by absence of growth period and a short fattening duration, with a wrapped haylage or corn silage-based diet. The second management was characterized by short pre-weaning and growth periods, and a long fattening period.

## 1. Introduction

In 2020, the European Union (EU) was the third-highest beef producer in the world, and France was the highest EU producer of beef, with 18% of EU production [1]. In France, young bulls were the second-highest fattened animal type (31%) after cull cows (47%) [1]. In 2021, 54% of the young bulls produced in France were exported as live fattened animals (2%) or meat (52%), to other countries in the EU (e.g., Italy, Germany, and Greece) and to Maghreb [2]. The carcass and meat quality of young bulls or steers have been extensively studied in the literature. Many factors have been identified as affecting carcass and meat quality throughout the continuum of farm to fork (e.g., rearing factors, stress, and aging). Cattle breeding is the first step which influences the final quality of the products (carcass and meat). It is in the interests of the beef sector stakeholders to manage, at an early stage, the potential quality of the live cattle. In regard to young bulls, many works have studied the individual effects of various rearing factors (e.g., age at weaning, growth rate, slaughter age, diet, fattening duration) on carcass and beef meat quality [3,4,5,6,7]. However, these results were mainly related to the fattening period, while the previous periods of the life of young bulls have been little studied. In heifers, recent studies have shown that rearing management (RM)—a combination of many rearing factors, applied throughout their life—contribute to the variability observed in their carcass and meat quality [8,9]. As previously observed for heifers, we supposed that different life periods could have an impact on the carcasses and meat quality of young bulls, and that it was possible to produce carcass and meat with similar properties, from different RMs. Several rearing factors within a life period may interact. The aims of this work were to characterize different RMs, and to study their effects on carcass and meat quality. As the carcass is composed of many muscles, the meat analyses of this study were mainly performed on the same muscle (longissimus muscle, LM), to compare our results with other studies. This muscle is the most studied in beef quality studies. The applied aim was to provide advice to the stakeholders in the beef chain, to allow the production of fattened young bulls according to quality traits.

## 2. Materials and Methods

### 2.1. Animals and Rearing Factors

The individual data of 179 Charolais young bulls, from 15 French commercial farms in 3 departments (Allier, Loire, and Puy-de-Dôme) of the Auvergne–Rhône–Alpes area, were used in this study. These farms were selected for their high diversity of RMs. The young bulls were born between October 2017 and March 2019, and slaughtered between April 2019 and July 2020. The life of the young bulls was divided into 3 key periods: pre-weaning period (birth to weaning, PWP), growth period (weaning to beginning of fattening, GP), and fattening period (beginning of fattening to slaughter, FP).

### 2.2. Rearing Factors

To collect the information on the rearing factors applied by the farmers throughout the young bulls’ life, a face-to-face survey was carried out, using questionnaires, and establishing batch management practices with each farmer [10]. From the batch management established, the duration and the ages were calculated for each housing and outside period of each key period considered. The average diet intake was calculated for the housing period, the outside period, and the whole period of each key period PWP, GP, and FP, respectively, according to Soulat et al. [11]. Each mean diet was characterized by the proportion of each forage, the concentrate quantity, and the average concentrate’s crude protein (CP) and net energy (NE). For the concentrates used in the different diets, the composition and the nutritional values of the purchased concentrates were collected from the manufacturers, and for the concentrates produced on the farm (e.g., barley, wheat, corn), the nutritional values of the INRAE system were used [12].

Finally, 30 rearing factors (quantitative or qualitative) were obtained and considered in this study. The rearing factors are defined in Table 1, with 18, 5, and 7 rearing factors related to PWP, GP, and FP, respectively.

### 2.3. Slaughtering, Carcass Traits, and Sampling

The young bulls were slaughtered in three French industrial slaughterhouses (SICABA, Bourbon-l’Archambault; SICAREV, Roanne; and SOCOPA, Villefranche-d’Allier). The carcasses were weighted and graded (conformation and fat scores) visually by an official judge, according to the EUROP grid system [13]. For the EUROP conformation scores, a numerical conversion was performed, to obtain a scale with 15 levels, where 1 = P− to 15 = E+. Then, the carcasses were chilled and stored at 2 °C until 24 h post-mortem. Afterwards, the right-hand side of each carcass was cut at the 6th rib level, and trained slaughterhouse staff evaluated the 11 following carcass traits. Subcutaneous fat thickness assessment was performed, using a caliper in the area shown in Figure 1. Six carcass traits—LM seepage (‘1 = the cut section was dry with no drop’ to ‘5 = the cut section had important drop’); intermuscular fat (‘1 = limited development’ to ‘5 = large amount’); nerves (‘1 = lack of visible nerves’ to ‘5 = many visible nerves’), overall meat grain; LM meat grain; and rhomboideus (RH) meat grain (‘1 = smooth, soft, without harshness’ to ‘5 = very rough/granular’)—were assessed, using a scale from 1 to 5 with a step of 0.5 [14]. The homogeneous color between the ribs’ muscles was assessed with a scale from 1 to 4 (1 = homogeneous; 2 = bicolor; 3 = tricolor; and 4 = more than 3 colors). Fat color, longissimus muscle color, and LM marbling were assessed, using the fat and meat color reference standards and the marbling reference standards described by UNECE [15].

Finally, a sample compounded of two ribs (5th and 4th ribs, localized in the chuck sale section) were withdrawn from the right-hand side of each carcass, and deboned. Each sample was vacuum-packaged and aged for 14 days at 4 °C, then frozen at −20 °C until the analyses.

### 2.4. Meat Quality Evaluation

After thawing (around 48 h at 4 °C), each beef rib sample was dissected, to separate the LM and serratus ventralis (SV) muscles, by professional butchers (INRAE Unité Expérimentale Herbipôle, Theix, Brittany, France). Both muscles were chosen to characterize the rib, which is composed of different muscles [16]. Then, each muscle was individually vacuum-packaged. The LM samples were conserved at 4 °C, before the color and sensory analyses, and the SV samples were frozen at −20 °C, until the shear force analyses.

The color of the LM samples was analyzed three hours before the sensory analyses, using a spectrophotometer (Konica Minolta CR-400, Osaka, Japan) and the CIE L*a*b* units [17]. To characterize the color of the LM, 6 measurements (randomly distributed) were carried out on the same LM area that was used to assess the LM color in the slaughterhouse. Then, 2 or 3 steaks (2 cm thickness) were cut from each LM sample, and were used for the sensory analyses. The rest of each LM sample was individually vacuum-packaged, and frozen at −20 °C until the texture analyses.

During the sensory sessions, the steaks were cooked in an aluminum foil on a plancha at 300 °C, to reach an internal temperature of 55 °C. Then, they were cut into homogeneous pieces (size 15 mm × 20 mm × 20 mm) and kept warm. For each sensory session, 8 samples were evaluated monadically by 10 trained people, using a Latin square design. For this study, 50 persons were trained (six 1-hour training sessions, in accordance with ISO 8586-1 standard [18]) to evaluate 10 sensory descriptors and one hedonic descriptor (Table 2), using the Tastel software^®^ version 2019 (ABT Informatique, Rouvroy-sur-Marne, France). The 10 sensory descriptors were evaluated using a 10-cm unstructured scale (from ‘0 = no perception’ to ‘10 = very intense perception’), and the overall acceptability (hedonic descriptor) used a scale anchoring (from ‘0 = I don’t like at all’ to ‘10 = I like very much’).

The LM samples used for the texture profile analysis (TPA) were thawed for 25 min, and cut to obtain raw meat cylinders (1 cm thickness and 1 cm in diameter), using a cookie cutter. To finish their thawing (around 30 min), the raw meat cylinders were kept at 4 °C. Each raw meat cylinder underwent 2 cycles of 20% compression at 4 °C, using rheometer (Kinexus pro+, Malvern Instruments, Malvern, UK) and the rSpace 1.61 software (Kinexus, Malvern, UK). The force-deformation curve obtained during TPA allowed for the calculation of six parameters: springiness; hardness; cohesiveness; resilience; gumminess; and chewiness [19,20]. A sample was not analyzed if it was not possible to cut at least 2 cylinders.

The SV samples were thawed at 4 °C for about 24 h, before the shear force analysis was carried out. From each raw SV sample, around 14 meat portions (1 cm wide, and between 0.9 and 1.1 cm thickness) were cut in parallel to the fibers. For each meat portion, in a perpendicular direction to the fibers, 2 shear force measurements were performed, using the Warner–Braztler method (Instron 5944, Elancourt, France) and Bluehill 2 software (Instron, Elancourt, France).

### 2.5. Statistical Analyses

Statistical analyses were performed using R 4.0.5 software (R core Team, Vienna, Austria) [21].

Firstly, a descriptive analysis of the rearing factors was realized (e.g., graphic distribution and quantile–quantile plots). According to the distribution of the rearing factors’ values, some quantitative rearing factors were converted into qualitative rearing factors, as described by Soulat et al. [9].

Then, the RMs applied throughout the life of the young bulls were defined from all rearing factors (q = 30, Table 1), using the factor analysis for mixed data (FAMD), followed by a hierarchical clustering on principal components (HCPC). The number of RMs considered in this study was determined from the dendrogram of the HCPC. The FAMD and HCPC were performed using the ‘FactoMineR’ package in R [22]. To characterize each RM, ANOVA and Khi2 were carried out on the quantitative and the qualitative rearing factors, respectively, as described by Soulat et al. [9].

Finally, ANOVA were performed, to evaluate the effect of the RMs on each carcass and meat traits.

In the ANOVA, the slaughterhouse and the operator effects were tested for all carcass traits. If these effects were significant, they were considered as random effects in a new ANOVA analysis (mixed model). However, if they were not significant, a new ANOVA was performed without considering these effects. As described by Soulat et al. [9], mixed models were performed for the sensory data, considering the panelist effect as a random effect. After each ANOVA and mixed model analysis, a Tukey test was realized.

The ANOVA were carried out using the ‘agricolae’ package [23] and the mixed models, followed by Tukey tests using the ‘lmerTest’, ‘emmeans’, ‘multcompView’, and ‘multcomp’ packages [24,25,26,27].

## 3. Results and Discussion

### 3.1. Characterization of the Rearing Managements

The FAMD and HCPC analysis defined five RMs from all rearing factors applied throughout the life of the young bulls (Table 3, Table 4 and Table 5). The RMs are described in Appendix A, and summarized in Figure 2. In summary, the young bulls from RM-1 had the shortest period with concentrates in their diet before their weaning, and were weaned the oldest. In RM-1, there was no GP, and the young bulls had the shortest fattening duration, with a diet base on wrapped haylage or corn silage. In RM-2, there was an important part of artificial insemination, and the majority of the calvings were assisted. Before weaning, the young bulls had the shortest outside-period duration. After weaning, the young bulls received a diet based on wrapped haylage and corn silage during GP and FP, respectively, and consumed a high concentrate quantity. Short PWP and GP durations characterized RM-3. During GP, the young bulls consumed a hay-based diet with a weak concentrate quantity. The young bulls had the longest fattening duration, and were young at the beginning of the fattening. Before weaning, the young bulls from RM-4 had only their mothers’ milk during the housing period, and had a long pasture-period duration. During GP, the young bulls received a diet based on corn silage or & straw, and consumed between 200 and 300 kg of concentrates. In RM-4, the young bulls were young at the beginning of FP, and received a straw-based diet. In RM-5, the young bulls had a very short period in housing, and a long period at pasture, before their weaning. The young bulls had the longest GP, and received a grass silage-based diet with between 300 and 600 kg of concentrates. During FP, the young bulls were the oldest at the beginning, and received a hay-based diet.

### 3.2. Effect of Rearing Managements on Carcass Traits

The RMs had an effect mainly on carcass traits related to fat and overall meat grain (Table 6).

The fat score of carcasses from RM-2 were significantly lower than the four other RMs. The subcutaneous fat thickness of carcasses was higher for heifers from RM-5 compared to those from RM-2, RM-3, and RM-4. In the literature, very few works have studied the effect of the periods before the fattening, on the carcass traits. Although the young bulls were weaned at different ages in this study, Blanco et al. [3] did not show a significant effect of the weaning age on the carcass fat score. Concerning FP, slaughter age and the fattening diet are the main rearing factors described in the literature as having an effect on the carcass fat score. Many studies observed that the carcass had a higher subcutaneous fat thickness and fat score when the young bulls or steers were slaughtered at an older age [4,28]. However, Do Prado et al. [29] did not observe any effect of the slaughter age (16 vs. 22 months), in young bulls, on the fat proportion in the carcass. Although the young bulls from RM-1 and RM-4 were slaughtered younger than those from RM-5 (Table 5), the fat scores were not significantly different (Table 6). These results were therefore close to those of Do Prado et al. [29]. The fattening diets were different between the five RMs characterized in this study. Many studies did not observe an effect of the fattening diet on the carcass fat score and/or the fat thickness [30,31], in young bulls or steers. However, Warren et al. [32] observed higher fat score for carcasses with a grass silage-based diet than those with concentrate-based diet, in steers. Cerdeño et al. [5] observed lesser thickness of the subcutaneous fat, and lower fat score, for young bulls receiving an alfalfa hay + concentrates diet, than for those receiving a straw + concentrate diet. The study of the animal’s whole life makes it difficult to compare the results obtained in this study with those previously published. However, this study confirmed that the differences observed for each carcass trait could not be explain only by the fattening diet of the young bulls.

At the 6th rib level, the intermuscular fat was significantly lower for carcasses from RM-1 and RM-2 than for carcasses from RM-4 and RM-5. As for the previous carcass traits, in the literature, it was the same rearing factors which were studied on the intermuscular fat [4,5,28,31]. When young bulls or steers were slaughtered at an older age, Marti et al. [28] and Bures and Barton [4] observed more intermuscular fat. However, the results obtained showed that the intermuscular fat of the carcass was similar for different slaughter ages (RM-4 = 16.2 months vs. RM-5 = 20.3 months) as observed by Aydin et al. [33], over LM. Other studies had observed the effect of the fattening duration on the intermuscular fat. Oezluetuerk et al. [7] did not observe any effect of the fattening duration on the fat thickness over LM, in young bulls. Our study confirmed these results. However, Keane et al. [34] observed that an increase of the fattening duration (105 vs. 175 days) increased the rib intermuscular fat. The effect of diet composition on the intermuscular fat was also studied. The production system before the FP [35] and the fattening diet [31] had no significant effect on the intermuscular fat at the level of the rib or the leg, in young bulls or steers. However, Keane et al. [34] observed an increase of rib intermuscular fat in steers fed a fattening diet composed of ad libithum concentrate and restricted grass silage, compared to a diet composed of grass silage with or without concentrate.

Marbling was not significantly different between the RMs, which differed by, e.g., the fattening diet, the slaughter age, and the fattening duration. The lack of effect of the fattening duration [7] and fattening diet [36,37] on the marbling were confirmed. However, Marti et al. [28] and Costa et al. [31] observed an increase of intramuscular fat when the animals were slaughtered at an older age, and when the Barrosã bulls received a low percentage of corn silage (30% vs. 70%) in their diet. Costa et al. [31] did not observe this effect with the Alentejana bulls.

The carcasses of young bulls from RM-4 had whiter fat than the carcasses from the other RMs. According to the fat color chart used, although there was a significant difference between RMs in regard to fat color, there was very little difference between a score of 0 and 1. In RM-4, the young bulls were weaned at the youngest age, and had a long pasture duration before their weaning. The weaning age in RM-4 could explain the lowest fat color score. However, Blanco et al. [3] did not observe a significant effect of the weaning age on the yellowness of the fat. The fat color may have been impacted by other rearing factors, such as diet in relation to the forage carotenoid content [38].

The color at the 6th rib level was more heterogeneous for carcasses from RM-4 than those from RM-2. According to the color chart used [15], RM-1, RM-2, RM-3, and RM-5 had tendencies to produce carcasses with a slightly darker LM than RM-4. According to the literature, at 24 h post mortem, the redness (a*) of the LM was not significantly impacted by the weaning age [3], the fattening diet [39], or the slaughter age [4]. However, Marti et al. [28] observed that the meat was darker (lower a* value) when young bulls or steers were slaughtered at 14 months, than when they were slaughtered at 10 or 12 months.

The overall meat grain was significantly smoother in young bulls from RM-1 and RM-2 than in those from RM-4. However, the LM and RH meat grains were not significantly different according to the RMs. To our knowledge, the RM effect on the meat grain has not been studied previously in young bulls.

The young bulls’ RM also had no significant effect on the weight and the conformation of the carcass.

In France, the meat market is mainly looking for carcasses with a fat score of 3, according to the EUROP grill. This target can be reached from RM-1, RM-3, RM-4, and RM-5. According to the carcass properties for other interesting traits for the beef industry, some RMs could be favored. For example, RM-1, RM-2, and RM-3 would be favored, if the meat industry target was to produce carcasses with low intermuscular and subcutaneous fat, a smooth meat grain, and a homogeneous meat color.

### 3.3. Effect of Rearing Managements on Meat Traits

The RMs had very few effects on the meat properties (Table 7). The RMs had a significant impact only on the color and the tenderness of the meat.

The raw LM meat had significantly higher lightness (L*) and yellowness (b*) values when RM-4 was applied, compared to RM-1, RM-2, and RM-3. The meat from RM-2 and RM-4 had a tendency to be redder. As observed for the carcass, the previous published works mainly studied the effects of rearing factors related to FP on the color of aging meat.

Many studies have observed no effect of the fattening diet on the three color parameters (L*a*b*) in young bulls after aging [31,39]. However, Moloney et al. [40] and Keady et al. [41] observed that the LM was redder and yellower when the steers received a concentrate-based diet or a high concentrate quantity (3 vs. 5 kg), respectively. Pesonen et al. [42] observed an increase of lightness when the young bulls received more concentrate (200 vs. 500 g/kg DM). The fattening diet of RM-4 was a straw-based diet, compared to RM-1 and RM-2 (Table 5), with no significant differences between the concentrate quantities in the diets of these three RMs (RM-1, RM-2, and RM-4). Moreover, although the young bulls from RM-5 were slaughtered at the oldest age, the meat produced was not significantly lighter or yellower compared to the meat from the other RMs. These results did not confirm those of Sargentini et al. [43], which showed that young bulls slaughtered at 24 months produced a lighter and yellower LM meat than young bulls slaughtered at 18 months. This difference could be explained by the fact that all young bulls were slaughtered before 22 months, in our study.

The cooked LM meat was significantly more tender (initial and overall) in the young bulls from RM-3 and RM-4 than in those from RM-2 (Table 7). Similar tenderness of the LM meat was obtained from RM-1, RM-3, RM-4, and RM-5. For aging duration, similarly to our study, many previous studies observed no effects of the fattening diet or the slaughter age on the LM tenderness (sensory or rheological assessments) [43,44,45]. However, Moloney et al. [40] observed that the meat was tougher (high shear force value) when the steers received a fattening diet with only concentrates, compared to those receiving grass silage during this period. Contrary to Moloney et al. [40], the LM tenderness from RM-4 (straw + concentrate fattening-based diet) was not significantly different to the other RMs using other fattening diets. Bures and Barton [4] showed that the LM meat was more tender when the young bulls were slaughtered at 18 months instead of 14 months.

The young bulls from RM-2 and RM-3 had tendencies to produce a raw SV muscle more tender than those from RM-1, RM-4, and RM-5 (Table 7). To our knowledge, the effect of the RMs on the shear force of raw SV has not been previously studied. The raw LM meat had a tendency to be more cohesive, resilient, gummy, and chewy when RM-1 and RM-2 were applied during the life of young bulls.

The young bulls from RM-1 had a tendency to produce LM meat with a higher fat aroma than the other RMs. To our knowledge, the effect of the RMs on the fat aroma has not been studied previously. For the LM meat, the atypical flavor had a tendency to be higher from RM-4, and lower from RM-1 and RM-2. The RMs had no significant effect on the other sensory descriptors and the hedonic descriptors. For an aging of 14 days, many studies showed no effect of the fattening diet and the slaughter age on the juiciness, the flavor intensity, and the overall acceptability [45,46]. However, Kerth et al. [44] observed that the LM meat was juicier and had higher flavor intensity when the steers received a concentrate-based diet, compared to those having a pasture period before receiving the same concentrate-based diet. Bures and Barton [4] showed that the LM meat (aged to 11 days) had high flavor intensity and overall acceptability, and low juiciness when the young bulls were slaughtered at 18 months instead of 14 months.

To produce the more tender LM meat (raw or cooked), RM-1, RM-3, RM-4, and RM-5 seem the most interesting for the young bulls. Nevertheless, in Europe, the classification system used to determine the sale price is based on carcass traits (weight, conformation, and fat scores). As a result, the interests of the farmers do not take into consideration the meat properties, a fact which can obstruct the enhancement of the meat quality. With the aim of jointly improving the carcass and meat quality, this study showed that RM-1 and RM-3 seem to be the best trade-off to achieve this aim. RM-1 and RM-3 allow for the production of high-quality carcasses and more tender LM meat.

## 4. Conclusions

For the first time, the effects of RMs applied throughout the life of young bulls on carcass and meat quality, were studied jointly. This study showed that the different RMs induced more variability in the carcass traits than in those of the meat. The RMs mainly impacted the traits related to fat and the overall meat grain of the carcass, the color (lightness and yellowness), and the tenderness of LM meat. The other carcass and meat traits studied were not significantly impacted by the RMs. This study also showed that it is possible to produce carcasses or meat with similar properties from different RMs. The best trade-off, to produce high carcass and meat quality jointly, was reached in young bulls from RM-1 or RM-3. RM-1 was mainly characterized by no GP and a short fattening duration with wrapped haylage-based or corn silage-based diets. RM-3 was mainly characterized by short PWP and GP, and a long FP. It would be interesting to complement these results by studying the effects of RMs on the quality of other muscles, so as to assess the overall quality of the carcass and the meat.

## Figures and Tables

**Figure 1 foods-11-02878-f001:**
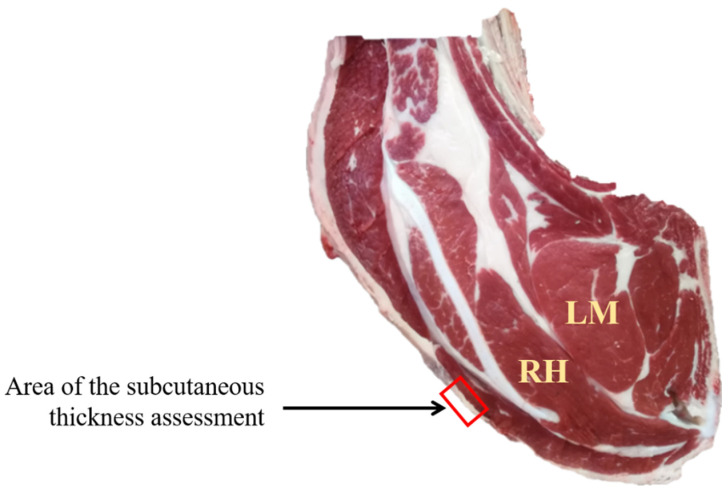
Localization of the longissimus and rhomboideus muscles (LM and RH, respectively) and the area of the subcutaneous fat assessment.

**Figure 2 foods-11-02878-f002:**
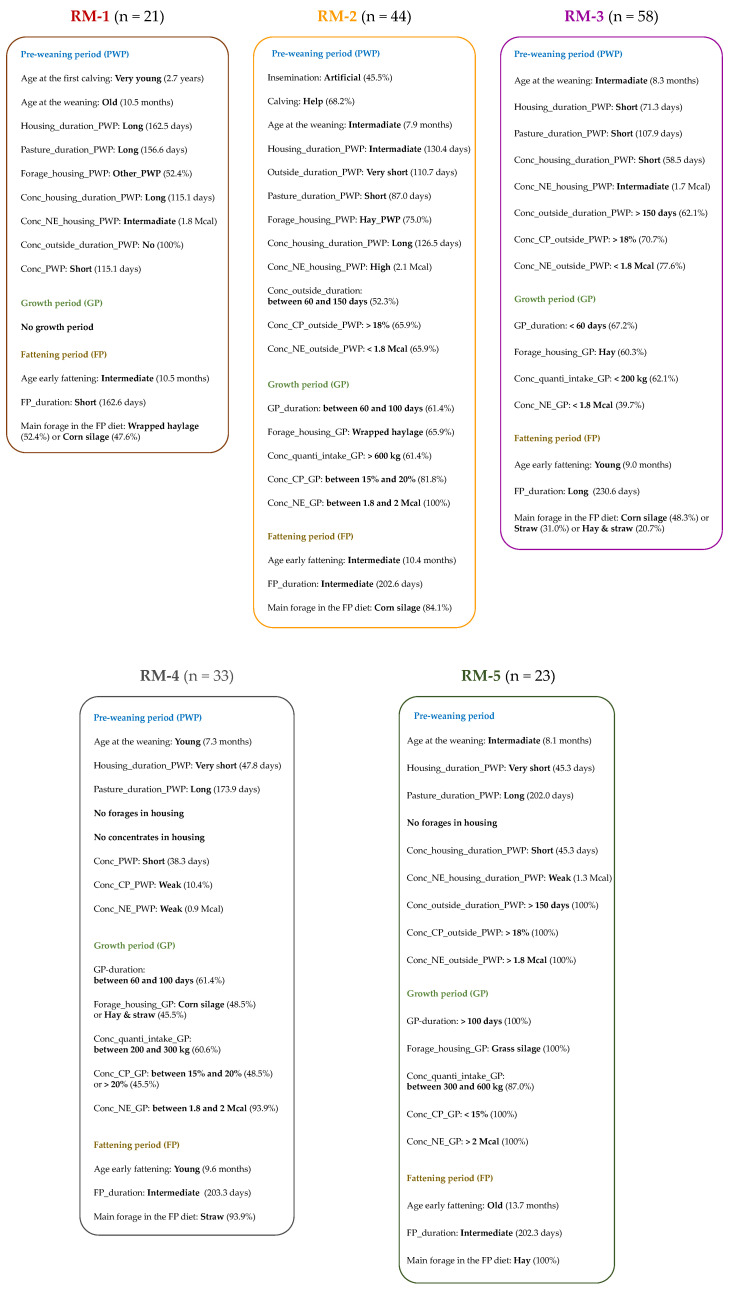
Description of the five rearing managements (RMs) applied throughout young bulls’ life, with a focus on the rearing factors characterizing the most each RM, as described in Table 3, Table 4 and Table 5.

**Table 1 foods-11-02878-t001:** Definition of the rearing factors characterizing each key period of the young bull’s life.

**Pre-Weaning Period**		
**Quantitative Rearing Factors**	**Description of the Rearing Factor**	
Age of the cow (years)	Age of the young bull’s mother at the young bull’s birth.	
Age at first calving (years)	Age of the young bull’s mother at first calving.	
Age at weaning (months)	Age of the young bull at weaning.	
Housing_duration_PWP (days)	Number of days spent in stall during PWP.	
Outside_duration_PWP (days)	Number of days spent outside during PWP.	
Pasture_duration_PWP (days)	Number of days spent at pasture during PWP.	
Conc_housing_PWP (days)	Number of days of offered concentrates in the calf’s diet during housing.	
Conc_PWP (days)	Number of days of offered concentrates in the calf’s diet during PWP.	
Conc_CP_housing_PWP (%)	Calculated average of the concentrates’ crude protein in the housing diet during PWP.	
Conc_NE_housing_PWP (Mcal)	Calculated average of the concentrates’ net energy in the housing diet during PWP.	
Conc_CP_PWP (%)	Calculated average of the concentrates’ crude protein in the diet during PWP.	
Conc_NE_PWP (Mcal)	Calculated average of the concentrates’ net energy in the diet during PWP.	
**Qualitative Rearing Factors**	**Modalities of the Rearing Factors**	**Description of the Rearing Factor**
Insemination type	Artificial	Artificial insemination using frozen semen.
	Natural	Insemination performed by a bull.
Calving	Easy	Natural calving.
	Help	Farmer intervention during the calving.
Forage_housing_PWP	Other_PWP	The main forage in the housing diet of the calf during PWP was grass silage or corn silage or wrapped haylage or alfalfa hay.
	Hay_PWP	Hay was the only forage in the housing diet of the calf during PWP.
	No	No offered forages in housing calf diet during PWP.
Conc_outside_duration_PWP	No	No offered concentrates in outside calf diet during PWP.
	[60 days; 150 days]	Offered concentrates in outside calf diet during PWP between 60 and 150 days.
	>150 days	Offered concentrates in outside calf diet during PWP above 150 days.
Conc_CP_outside_PWP (%)	No	No offered concentrates in pasture calf diet during PWP.
	<18%	Across the whole pasture of PWP, the calculated average of the concentrates’ crude protein content was below 18%.
	>18%	Across the whole pasture of PWP, the calculated average of the concentrates’ crude protein content was above 18%.
Conc_NE_outside_PWP (Mcal)	No	No offered concentrates in pasture calf diet during PWP.
	<1.8 Mcal	Across the whole pasture of PWP, the calculated average of the concentrates’ net energy content was below 1.8 Mcal.
	>1.8 Mcal	Across the whole pasture of PWP, the calculated average of the concentrates’ net energy content was above 1.8 Mcal.
**Growth Period**		
**Qualitative Rearing Factors**	**Modalities of the Rearing Factors**	**Description of the Rearing Factor**
GP_duration (days)	No	The young bulls had no GP.
	<60 days	The GP duration was below 60 days.
	[60 days; 100 days]	The GP duration was between 60 and 100 days.
	>100 days	The GP duration was above 100 days.
Forage_housing_GP	No	The young bulls had no GP.
	Wrapped_haylage	Across the whole GP, the main forage in the housing diet was wrapped haylage (>80%).
	Corn_silage	Across the whole GP, the main forage in the housing diet was corn silage (>57%).
	Hay_GP	Across the whole GP, the main forage in the housing diet was hay (>75%).
	Straw_GP	Across the whole GP, the main forage in the housing diet was straw (>60%).
	Grass_silage_GP	Across the whole GP, the main forage in the housing diet was grass silage (>65%).
	Hay and Straw_GP	Across the whole GP, the main forage in the housing diet was hay and straw (100%).
Conc_quanti_intake_GP (kg)	No	The young bulls had no GP.
	<200 kg	The concentrate quantity intake during the GP was below 200 kg.
	[200 kg; 300 kg]	The concentrate quantity intake during the GP was between 300 and 600 kg.
	[300 kg; 600 kg]	The concentrate quantity intake during the GP was between 200 and 300 kg.
	>600 kg	The concentrate quantity intake during the GP was above 600 kg.
Conc_CP_GP (%)	No	The young bulls had no GP.
	<15%	Across the whole GP, the calculated average of the concentrates’ crude protein content was below 15%.
	[15%; 20%]	Across the whole GP, the calculated average of the concentrates’ crude protein content was between 15% and 20%.
	>20%	Across the whole GP, the calculated average of the concentrates’ crude protein content was above 20%.
Conc_NE_GP (Mcal)	No	The young bulls had no GP.
	<1.8 Mcal	Across the whole GP, the calculated average of the concentrates’ net energy content was below 1.8 Mcal.
	[1.8 Mcal; 2 Mcal]	Across the whole GP, the calculated average of the concentrates’ net energy content was between 1.8 Mcal and 2 Mcal.
	>2 Mcal	Across the whole GP, the calculated average of the concentrates’ net energy content was above 2 Mcal.
**Fattening Period**		
**Quantitative Rearing Factors**	**Description of the Rearing Factor**	
Age of early fattening (months)	Age of the young bull at the beginning of FP.	
Slaughter age (months)	Age of the young bull at slaughter.	
FP_duration (days)	Number of days between the beginning of FP and slaughter.	
Conc_quanti_intake_FP (kg)	Total concentrate quantity intake per heifer during the whole FP.	
Conc_CP_FP (%)	Calculated average of the concentrates’ crude protein content across the whole FP.	
Conc_NE_FP (Mcal)	Calculated average of the concentrates’ net energy content across the whole FP.	
**Qualitative Rearing Factors**	**Modalities of the** **Rearing Factors**	**Description of the Rearing Factor**
Main forage in the FP diet (%)	Wrapped_haylage_FP	The percentage of wrapped haylage in the FP diet was 100%.
	Corn_silage_FP	The percentage of corn silage in the FP diet was above 76%.
	Hay_FP	The percentage of hay in the FP diet was above 80%.
	Straw_FP	The percentage of straw in the FP diet was above 75%.
	Hay&Straw_FP	The percentage of hay and straw in the FP diet was 100%.

**Table 2 foods-11-02878-t002:** Definitions of the sensory and hedonic descriptors.

Descriptors	Definition
Red color intensity	Refers to the red color intensity of the meat sample after cooking (‘0 = light’ to ‘10 = dark’).
Initial tenderness	Facility to chew and cut the meat sample at the first bite (‘0 = tough’ to ‘10 = very tender’).
Overall tenderness	Time of chewing and number of chews required to masticate the meat sample, ready for swallowing (‘0 = tough’ to ‘10 = very tender’).
Overall juiciness	Perception of water content in the meat sample during the mastication (‘0 = dry’ to ‘10 = very juicy’).
Presence of nerves	Quantity of nerves perceived in the meat sample (‘0 = none’ to ‘10 = very important’).
Residue	Amount of residue after chewing (‘0 = none’ to ‘10 = very important’).
Flavor intensity	Global flavor intensity assessment of the beef meat (‘0 = none’ to ‘10 = very intense’).
Fat aroma	Fat aroma intensity (‘0 = none’ to ‘10 = very intense’).
Atypical flavor	Flavor associated with aromas that should not normally be present in meat (e.g., aftertaste, rancid) (‘0 = none’ to ‘10 = very intense’).
Flavor persistence	Refers to remnant beef flavor duration in the mouth, perceived after swallowing (‘0 = very quick’ to ‘10 = very long’).
Overall acceptability	Overall liking (hedonic perception) of the meat sample (‘0 = highly disliked’ to ‘10 = highly liked’).

**Table 3 foods-11-02878-t003:** Rearing factors characterizing the pre-weaning period (PWP) of the rearing managements (RMs) applied throughout the whole life of the young bulls.

Pre-Weaning Period		Overall (*n* = 179)	Rearing Managements					*p*
			RM-1 (*n* = 21)	RM-2 (*n* = 44)	RM-3 (*n* = 58)	RM-4 (*n* = 33)	RM-5 (*n* = 23)	
Quantitative Rearing Factors		Mean ± SE	Mean ± SE	Mean ± SE	Mean ± SE	Mean ± SE	Mean ± SE	
Age of the cow (years)		5.3 ± 0.1	5.1 ± 0.5	5.0 ± 0.3	5.8 ± 0.3	5.0 ± 0.4	5.0 ± 0.4	0.16
Age at first calving (years)		2.9 ± 0.01	2.7 ^c^ ± 0.06	2.9 ^b^ ± 0.03	3.0 ^a^ ± 0.03	3.0 ^ab^ ± 0.02	2.9 ^ab^ ± 0.02	<0.001
Age at weaning (months)		8.3 ± 0.1	10.5 ^a^ ± 0.3	7.9 ^b^ ± 0.1	8.3 ^b^ ± 0.09	7.3 ^c^ ± 0.1	8.1 ^b^ ± 0.2	<0.001
Housing_duration_PWP (days)		88.8 ± 4.1	162.5 ^a^ ± 2.4	130.4 ^b^ ± 7.2	71.3 ^c^ ± 4.2	47.8 ^d^ ± 6.6	45.3 ^d^ ± 5.7	<0.001
Outside_duration_PWP (days)		162.4 ± 3.4	156.6 ^c^ ± 7.1	110.7 ^d^ ± 7.0	181.5 ^ab^ ± 3.7	173.9 ^bc^ ± 4.9	202.0 ^a^ ± 2.3	<0.001
Pasture_duration_PWP (days)		132.7 ± 5.7	156.6 ^a^ ± 7.1	87.0 ^b^ ± 10.6	107.9 ^b^ ± 11.7	173.9 ^a^ ± 4.9	202.0 ^a^ ± 2.3	<0.001
Conc_housing_PWP (days)		69.4 ± 4.1	115.1 ^a^ ± 8.9	126.5 ^a^ ± 6.6	58.5 ^b^ ± 3.1	0.0 ^c^ ± 0.0	45.3 ^b^ ± 5.7	<0.001
Conc_PWP (days)		167.2 ± 6.8	115.1 ^c^ ± 8.9	213.5 ^ab^ ± 6.5	192.5 ^b^ ± 11.5	38.3 ^d^ ± 7.0	247.4 ^a^ ± 5.0	<0.001
Conc_CP_housing_PWP (%)		13.1 ± 0.5	17.0 ^a^ ± 0.5	15.6 ^ab^ ± 0.4	16.9 ^a^ ± 0.4	0.0 ^c^ ± 0.0	13.9 ^b^ ± 1.1	<0.001
Conc_NE_housing_PWP (Mcal)		1.4 ± 0.06	1.8 ^b^ ± 0.002	2.1 ^a^ ± 0.07	1.7 ^b^ ± 0.05	0.0 ^d^ ± 0.0	1.3 ^c^ ± 0.1	<0.001
Conc_CP_PWP (%)		15.9 ± 0.4	17.0 ^a^ ± 0.5	17.3 ^a^ ± 0.3	17.5 ^a^ ± 0.1	10.4 ^b^ ± 1.9	15.7 ^a^ ± 0.001	<0.001
Conc_NE_PWP (Mcal)		1.7 ± 0.04	1.8 ^a^ ± 0.002	2.0 ^a^ ± 0.07	1.7 ^a^ ± 0.03	0.9 ^b^ ± 0.2	1.9 ^a^ ± 0.001	<0.001
**Qualitative Rearing Factors**	**Modalities of the Rearing Factors**	** *n* **						
Insemination type	Artificial	43	33.3%	45.5%	5.2%	39.4%	0%	<0.001
	Natural	136	66.7%	54.5%	94.8%	60.6%	100%	
Calving	Easy	125	66.7%	31.8%	84.5%	87.9%	82.6%	<0.001
	Help	54	33.3%	68.2%	15.5%	12.1%	17.4%	
Forage_housing_PWP	Other_PWP	27	52.4%	25.0%	8.6%	0%	0%	<0.001
	Hay_PWP	72	47.6%	75.0%	50.0%	0%	0%	
	No	80	0%	0%	41.4%	100%	100%	
Conc_outside_duration_PWP (days)	No	66	100%	34.1%	22.4%	51.5%	0%	<0.001
	[60 days; 150 days]	48	0%	52.3%	15.5%	48.5%	0%	
	>150 days	65	0%	13.6%	62.1%	0%	100%	
Conc_CP_outside_PWP (%)	No	66	100%	34.1%	22.4%	51.5%	0%	<0.001
	<18%	64	0%	0%	70.7%	0%	100%	
	>18%	49	0%	65.9%	6.9%	48.5%	0%	
Conc_NE_outside_PWP (Mcal)	No	66	100%	34.1%	22.4%	51.5%	0%	<0.001
	<1.8 Mcal	74	0%	65.9%	77.6%	0%	0%	
	>1.8 Mcal	39	0%	0%	0%	48.5%	100%	

Age of the cow: age of the young bull’s mother at the young bull’s birth. Age at first calving: age of the young bull’s mother at first calving. Age at weaning: age of the young bull at weaning. Housing_duration_PWP: numbers of days spent in stall during PWP. Outside_duration_PWP: number of days spent outside during PWP. Pasture_duration_PWP: number of days spent at pasture during PWP. Conc_housing_PWP: number of days of offered concentrates in the calf’s diet during housing. Conc_PWP: number of days of offered concentrates in the calf’s diet during PWP. Conc_CP_housing_PWP: calculated average of the concentrates’ crude protein in the housing diet during PWP. Conc_NE_housing_PWP: calculated average of the concentrates’ net energy in the housing diet during PWP. Conc_CP_PWP: calculated average of the concentrates’ crude protein in the diet during PWP. Conc_NE_PWP: calculated average of the concentrates’ net energy in the diet during PWP. Insemination type: artificial (artificial insemination using frozen semen); natural (insemination performed by a bull). Calving: easy (natural calving); help (farmer intervention during the calving). Forage_housing_PWP: other_PWP (the main forage in the housing diet of the calf during PWP was grass silage or corn silage or wrapped haylage or alfalfa hay); hay_PWP (hay was the only forage in the housing diet of the calf during PWP); no (no offered forages in housing calf’s diet during PWP). Conc_outside_duration_PWP: number of days of offered concentrates in the calf’s diet during outside period. Conc_CP_outside_PWP: calculated average of the concentrates’ crude protein in the outside diet during PWP. Conc_NE_outside_PWP: calculated average of the concentrates’ net energy in the outside diet during PWP. SE: standard error. n: number of young bulls. Values followed by different letters (a, b, c) are significantly different from each other at *p* ≤ 0.05.

**Table 4 foods-11-02878-t004:** Rearing factors characterizing the growth period (GP) of the rearing managements (RMs) applied throughout the whole life of the young bulls.

Growth Period		Overall (*n* = 179)	Rearing Managements					*p*
			RM-1 (*n* = 21)	RM-2 (*n* = 44)	RM-3 (*n* = 58)	RM-4 (*n* = 33)	RM-5 (*n* = 23)	
Qualitative Rearing Factors	Modalities of the Rearing Factors	*n*						
GP_duration (days)	No	41	100%	0%	31.0%	6.1%	0%	<0.001
	<60 days	53	0%	20.5%	67.2%	15.2%	0%	
	[60 days; 100 days]	48	0%	61.4%	1.7%	60.6%	0%	
	>100 days	37	0%	18.2%	0%	18.2%	100%	
Forage_housing_GP	No	43	100%	4.5%	31.0%	6.1%	0%	<0.001
	Wrapped_haylage	29	0%	65.9%	0%	0%	0%	
	Corn_silage	24	0%	18.2%	0%	48.5%	0%	
	Hay_GP	37	0%	4.5%	60.3%	0%	0%	
	Straw_GP	3	0%	6.8%	0%	0%	0%	
	Grass_silage_GP	24	0%	0%	1.7%	0%	100%	
	Hay and Straw_GP	19	0%	0%	6.9%	45.5%	0%	
Conc_quanti_intake_GP (kg)	No	41	100%	0%	31.0%	6.1%	0%	<0.001
	<200 kg	44	0%	6.8%	62.1%	15.2%	0%	
	[200 kg; 300 kg]	37	0%	22.7%	6.9%	60.6%	13.0%	
	[300 kg; 600 kg]	30	0%	9.1%	0%	18.2%	87.0%	
	>600 kg	27	0%	61.4%	0%	0%	0%	
Conc_CP_GP (%)	No	41	100%	0%	31.0%	6.1%	0%	<0.001
	<15%	24	0%	0%	1.7%	0%	100%	
	[15%; 20%]	71	0%	81.8%	32.8%	48.5%	0%	
	>20%	43	0%	18.2%	34.5%	45.5%	0%	
Conc_NE_GP (Mcal)	No	41	100%	0%	31.0%	6.1%	0%	<0.001
	<1.8 Mcal	23	0%	0%	39.7%	0%	0%	
	[1.8 Mcal; 2 Mcal]	79	0%	100%	6.9%	93.9%	0%	
	>2 Mcal	36	0%	0%	22.4%	0%	100%	

GP_duration: number of days between the weaning and the beginning of FP. Forage_housing_GP: no (the young bulls had no GP); wrapped_haylage (across the whole GP, the main forage in the housing diet was wrapped haylage [>80%]); corn_silage (across the whole GP, the main forage in the housing diet was corn silage [>57%]); hay_GP (across the whole GP, the main forage in the housing diet was hay [>75%]); straw_GP (across the whole GP, the main forage in the housing diet was straw [>60%]); grass_silage_GP (across the whole GP, the main forage in the housing diet was grass silage [>65%]); hay and straw_GP (across the whole GP, the main forage in the housing diet was hay and straw [100%]). Conc_quanti_intake_GP: total concentrate quantity intake per heifer during the whole GP. Conc_CP_GP: calculated average of the concentrates’ crude protein in the diet during GP. Conc_NE_GP: calculated average of the concentrates’ net energy in the diet during GP. n: number of young bulls.

**Table 5 foods-11-02878-t005:** Rearing factors characterizing the fattening period (FP) of the rearing managements (RMs) applied throughout the whole life of the young bulls.

Fattening Period		Overall (*n* = 179)	Rearing Managements					*p*
			RM-1 (*n* = 21)	RM-2 (*n* = 44)	RM-3 (*n* = 58)	RM-4 (*n* = 33)	RM-5 (*n* = 23)	
Quantitative Rearing Factors		Mean ± SE	Mean ± SE	Mean ± SE	Mean ± SE	Mean ± SE	Mean ± SE	
Age of early fattening (months)		10.2 ± 0.1	10.5 ^b^ ± 0.3	10.4 ^b^ ± 0.2	9.0 ^c^ ± 0.1	9.6 ^c^ ± 0.2	13.7 ^a^ ± 0.2	<0.001
Slaughter age (months)		17.0 ± 0.1	15.8 ^c^ ± 0.2	17.0 ^b^ ± 0.2	16.5 ^bc^ ± 0.1	16.2 ^c^ ± 0.2	20.3 ^a^ ± 0.2	<0.001
FP_duration (days)		207.1 ± 3.4	162.6 ^c^ ± 11.9	202.6 ^b^ ± 8.3	230.6 ^a^ ± 4.6	203.3 ^b^ ± 5.4	202.3 ^b^ ± 3.3	<0.001
Conc_quanti_intake_FP (kg)		1449.3 ± 32.9	1355.9 ^b^ ± 118.9	1298.1 ^b^ ± 58.4	1641.7 ^a^ ± 70.3	1387.5 ^b^ ± 43.2	1427.3 ^ab^ ± 23.7	<0.001
Conc_CP_FP (%)		18.6 ± 0.3	16.7 ^cd^ ± 0.1	20.3 ^a^ ± 0.2	19.6 ^ab^ ± 0.6	18.3 ^bc^ ± 0.3	14.5 ^d^ ± 0.0	<0.001
Conc_NE_FP (Mcal)		1.9 ± 0.01	1.9 ^ab^ ± 0.02	1.9 ^a^ ± 0.01	1.9 ^b^ ± 0.03	1.8 ^b^ ± 0.02	2.0 ^a^ ± 0.0	<0.001
**Qualitative Rearing Factors**	**Modalities of the Rearing Factors**	** *n* **						
Main forage in the FP diet (%)	Wrapped_haylage_FP	11	52.4%	0%	0%	0%	0%	<0.001
	Corn_silage_FP	77	47.6%	84.1%	48.3%	6.1%	0%	
	Hay_FP	27	0%	9.1%	0%	0%	100%	
	Straw_FP	52	0%	6.8%	31%	93.9%	0%	
	Hay&Straw_FP	12	0%	0%	20.7%	0%	0%	

Age of early fattening: age of the young bull at the beginning of FP. Slaughter age: age of the young bull at slaughter. FP_duration: number of days between the beginning of FP and slaughter. Conc_quanti_intake_FP: total concentrate quantity intake per heifer during the whole FP. Conc_CP_FP: calculated average of concentrate’s crude protein content across the whole FP. Conc_NE_FP: calculated average of concentrate’s net energy content across the whole FP. Main forage in the FP diet: wrapped_haylage_FP (the percentage of wrapped haylage in the FP diet was 100%); corn_silage_FP (the percentage of corn silage in the FP diet was above 76%); hay_FP (the percentage of hay in the FP diet was above 80%); straw_FP (the percentage of straw in the FP diet was above 75%); hay&straw_FP (the percentage of hay and straw in the FP diet was 100%). SE: standard error. n: number of young bulls. Values followed by different letters (a, b, c) are significantly different from each other at *p* ≤ 0.05.

**Table 6 foods-11-02878-t006:** Effects of the five rearing managements (RM), applied throughout the young bulls’ life, on the carcass traits.

Carcass Traits	Overall	Rearing Managements					*p*
		RM-1	RM-2	RM-3	RM-4	RM-5	
	Mean ± SE	Mean ± SE	Mean ± SE	Mean ± SE	Mean ± SE	Mean ± SE	
	*n* = 179	*n* = 21	*n* = 44	*n* = 58	*n* = 33	*n* = 23	
Cold weight (kg)	444 ± 33	447 ± 4	439 ± 4	453 ± 5	439 ± 4	435 ± 5	0.15
Conformation score (scale 1 to 15)	10.2 ± 2.7	10.4 ± 0.3	10.1 ± 0.2	10.2 ± 0.2	9.9 ± 0.2	10.5 ± 0.3	0.16
Fat score (scale 1 to 5)	2.7 ± 0.2	2.8 ^a^ ± 0.1	2.4 ^b^ ± 0.1	2.8 ^a^ ± 0.1	2.8 ^a^ ± 0.1	3.0 ^a^ ± 0.1	<0.001
Assessment at the 6th rib level	*n* = 174	*n* = 21	*n* = 44	*n* = 55	*n* = 31	*n* = 23	
Subcutaneous fat (cm)	0.8 ± 0.06	0.5 ^ab^ ± 0.3	0.1 ^b^ ± 0.3	0.6 ^b^ ± 0.2	0.2 ^b^ ± 0.3	1.3 ^a^ ± 0.3	<0.001
Longissimus muscle seepage (scale 1 to 5)	2.2 ± 0.2	2.1 ± 0.3	2.1 ± 0.3	2.2 ± 0.3	2.6 ± 0.3	2.4 ± 0.4	0.28
Intermuscular fat (scale 1 to 5)	1.4 ± 0.1	1.1 ^c^ ± 0.1	1.1 ^c^ ± 0.1	1.3 ^bc^ ± 0.1	1.6 ^a^ ± 0.1	1.6 ^ab^ ± 0.1	<0.001
Nerves (scale 1 to 5)	1.5 ± 0.1	1.5 ± 0.2	1.2 ± 0.2	1.3 ± 0.2	1.2 ± 0.2	1.3 ± 0.2	0.48
Overall meat grain (scale 1 to 5)	2.0 ± 0.1	1.8 ^b^ ± 0.1	1.9 ^b^ ± 0.1	2.0 ^ab^ ± 0.1	2.4 ^a^ ± 0.1	2.2 ^ab^ ± 0.2	0.003
Longissimus meat grain (scale 1 to 5)	1.9 ± 0.1	1.8 ± 0.2	1.5 ± 0.2	1.9 ± 0.2	1.7 ± 0.2	1.7 ± 0.3	0.16
Rhomboideus meat grain (scale 1 to 5)	1.4 ± 0.1	1.2 ± 0.2	1.2 ± 0.2	1.2 ± 0.2	1.4 ± 0.2	1.4 ± 0.2	0.19
Fat color (scale 0 to 9)	1.5 ± 0.1	1.2 ^a^ ± 0.2	1.2 ^a^ ± 0.2	1.1 ^a^ ± 0.2	0.4 ^b^ ± 0.2	1.2 ^a^ ± 0.2	<0.001
Homogeneous color of muscles at the 6th rib (scale 1 to 4)	1.8 ± 0.1	1.9 ^ab^ ± 0.1	1.6 ^b^ ± 0.1	1.9 ^ab^ ± 0.1	2.3 ^a^ ± 0.1	1.9 ^ab^ ± 0.2	0.002
Longissimus color (scale 0 to 7)	2.4 ± 0.2	2.0 ± 0.4	2.0 ± 0.4	2.0 ± 0.3	1.5 ± 0.4	1.9 ± 0.4	0.08
Longissimus marbling (scale 0 to 6)	0.7 ± 0.05	0.8 ± 0.2	0.6 ± 0.2	0.6 ± 0.2	0.7 ± 0.2	0.5 ± 0.2	0.66

SE: standard error. n: number of young bulls. Values followed by different letters (a, b, c) are significantly different from each other at *p* ≤ 0.05.

**Table 7 foods-11-02878-t007:** Effects of the five rearing managements (RMs), applied throughout the young bulls’ life, on the meat traits.

Meat Traits	Overall	Rearing Managements					*p*
		RM-1	RM-2	RM-3	RM-4	RM-5	
	Mean ± SE	Mean ± SE	Mean ± SE	Mean ± SE	Mean ± SE	Mean ± SE	
Seratus ventralis muscle	*n* = 163	*n* = 20	*n* = 42	*n* = 55	*n* = 28	*n* = 18	
Shear force (N/cm^2^)	80.5 ± 6.3	83.4 ± 5.6	74.2 ± 3.6	76.9 ± 2.5	88.3 ± 5.6	85.0 ± 6.5	0.06
Longissimus muscle							
Raw meat							
Color descriptors	*n* = 166	*n* = 21	*n* = 42	*n* = 58	*n* = 27	*n* = 18	
L*	44.9 ± 3.5	44.5 ^b^ ± 0.5	43.9 ^b^ ± 0.5	44.3 ^b^ ± 0.4	47.3 ^a^ ± 0.6	46.2 ^ab^ ± 0.6	<0.001
a*	18.5 ± 1.4	17.1 ± 0.7	19.3 ± 0.5	18.4 ± 0.5	19.1 ± 0.6	17.5 ± 0.7	0.06
b*	12.2 ± 0.9	11.4 ^b^ ± 0.2	11.9 ^b^ ± 0.2	12.2 ^b^ ± 0.1	13.1 ^a^ ± 0.2	12.3 ^ab^ ± 0.2	<0.001
TPA texture profile	*n* = 126	*n* = 16	*n* = 30	*n* = 53	*n* = 17	*n* = 10	
Springiness	0.47 ± 0.04	0.4 ± 0.02	0.5 ± 0.01	0.5 ± 0.01	0.5 ± 0.02	0.4 ± 0.02	0.83
Hardness	1.5 ± 0.1	1.6 ± 0.1	1.5 ± 0.1	1.4 ± 0.05	1.5 ± 0.1	1.5 ± 0.1	0.52
Cohesiveness	2.3 ± 0.2	2.8 ± 0.4	2.7 ± 0.3	2.1 ± 0.2	1.8 ± 0.3	1.8 ± 0.3	0.06
Resilience	0.2 ± 0.02	0.3 ± 0.03	0.3 ± 0.02	0.2 ± 0.01	0.2 ± 0.02	0.2 ± 0.02	0.06
Gumminess	3.3 ± 0.3	4.3 ± 0.5	4.1 ± 0.5	3.0 ± 0.3	2.7 ± 0.5	2.7 ± 0.6	0.06
Chewiness	1.5 ± 0.1	1.9 ± 0.2	1.8 ± 0.2	1.3 ± 0.1	1.3 ± 0.2	1.3 ± 0.3	0.06
Cooked meat		Emmean ± SE	Emmean ± SE	Emmean ± SE	Emmean ± SE	Emmean ± SE	
Sensory descriptors (0–10 scale)	*n* = 166	*n* = 21	*n* = 42	*n* = 58	*n* = 27	*n* = 18	
Red color intensity	3.1 ± 0.2	3.2 ± 0.3	2.9 ± 0.2	3.3 ± 0.2	2.9 ± 0.3	2.6 ± 0.3	0.31
Initial tenderness	6.1 ± 0.5	6.1 ^ab^ ± 0.2	5.8 ^b^ ± 0.2	6.2 ^a^ ± 0.1	6.4 ^a^ ± 0.2	6.1 ^ab^ ± 0.2	0.001
Overall tenderness	5.7 ± 0.4	5.7 ^ab^ ± 0.2	5.4 ^b^ ± 0.2	5.8 ^a^ ± 0.2	5.9 ^a^ ± 0.2	5.7 ^ab^ ± 0.2	0.007
Overall juiciness	4.3 ± 0.3	4.6 ± 0.3	4.7 ± 0.3	4.4 ± 0.3	4.2 ± 0.3	4.4 ± 0.3	0.17
Presence of nerves	2.0 ± 0.2	1.9 ± 0.3	2.2 ± 0.2	2.0 ± 0.2	1.9 ± 0.2	2.0 ± 0.3	0.18
Residue	2.9 ± 0.2	3.0 ± 0.2	3.0 ± 0.2	2.9 ± 0.2	2.9 ± 0.2	3.0 ± 0.2	0.87
Flavor intensity	5.7 ± 0.4	5.7 ± 0.2	5.7 ± 0.2	5.7 ± 0.1	5.8 ± 0.2	5.7 ± 0.2	0.84
Fat aroma	3.5 ± 0.3	3.8 ± 0.3	3.4 ± 0.3	3.6 ± 0.3	3.6 ± 0.3	3.4 ± 0.3	0.10
Atypical flavor	1.1 ± 0.1	0.8 ± 0.3	0.8 ± 0.3	1.0 ± 0.3	1.3 ± 0.3	1.0 ± 0.3	0.07
Flavor persistence	4.7 ± 0.4	4.6 ± 0.2	4.7 ± 0.2	4.7 ± 0.2	4.8 ± 0.2	4.6 ± 0.2	0.79
Overall acceptability	5.1 ± 0.4	5.4 ± 0.2	5.2 ± 0.2	5.0 ± 0.2	5.2 ± 0.2	4.9 ± 0.2	0.41

SE: standard error. n: number of young bulls. Emmean: estimated marginal means. Values followed by different letters (a, b) are significantly different from each other at *p* ≤ 0.05.

## Data Availability

No new data were created or analyzed in this study. Data sharing is not applicable to this article.

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
