# Peer review of "Effects of Rearing Management Applied throughout the Charolais Young Bulls’ Life on Carcass and Meat Quality"

_foods, 2022, doi:10.3390/foods11182878_

Round 1

Reviewer 1 Report

Extensive research on interesting topics that are less frequently discussed. A synthetic introduction highlighting the importance of the analyzes performed.
Tables 1-3 very extensive, seem too rich in information. Perhaps they would be more readable if you removed "descriptions of the rearing factor" from them and put them under the tables. Materials and methods also described extensively and precisely.
Line 206, 284, 331, 332 - I propose continuity in the nomenclature of Rearing Managements: RM-5 not the five RM, for example.
Lines 216-218 - please provide source.
Very long list of References.

Wyniki tłumacExtensive research on less frequently discussed topics.

Author Response

Reviewer 1

Open Review

English language and style

( ) Extensive editing of English language and style required
( ) Moderate English changes required
( ) English language and style are fine/minor spell check required
(x) I don't feel qualified to judge about the English language and style

Yes

Can be improved

Must be improved

Not applicable

Does the introduction provide sufficient background and include all relevant references?

(x)

( )

( )

( )

Are all the cited references relevant to the research?

(x)

( )

( )

( )

Is the research design appropriate?

(x)

( )

( )

( )

Are the methods adequately described?

(x)

( )

( )

( )

Are the results clearly presented?

( )

(x)

( )

( )

Are the conclusions supported by the results?

(x)

( )

( )

( )

Comments and Suggestions for Authors

Extensive research on interesting topics that are less frequently discussed. A synthetic introduction highlighting the importance of the analyzes performed.
Thank for your comment

Tables 1-3 very extensive, seem too rich in information. Perhaps they would be more readable if you removed "descriptions of the rearing factor" from them and put them under the tables.

We have put under the tables 3-5 the descriptions of the rearing factors (L 207-217,220-225, 227-231). We had also a new table which described each rearing factors and presented the distribution.

We hope that this will allow a better understanding of the tables.

Materials and methods also described extensively and precisely.
Thank for your comment

Line 206, 284, 331, 332 - I propose continuity in the nomenclature of Rearing Managements: RM-5 not the five RM, for example.

We carried out modifications in the text to help the understanding (L 336, 337, 338, 352387, 389, 390). For example “five RM” did not refer to RM-5 but refer to the 5 RM considered in this study

Lines 216-218 - please provide source.

We added the source in the text (L 269)

Very long list of References.

It is true there are many references. However, in this study, there are many carcass and meat traits studied requiring many references to discuss our results.

Reviewer 2 Report

Dear Authors,

the submitted manuscript evaluates the effect of different management factors at all stages of rearing on the final quality of the carcasses and meat. While acknowledging the originality of the study, the manuscript needs a thorough review before publication.

The text refers to the need "to produce simultaneously high carcass and LM meat qualities", this concept needs improvement; First of all, I believe that the aim is not to produce high quality LM meat, but to produce high quality meat. The investigations are carried out on LM because the characteristics of the muscle allow the tests to be repeatable and comparable.

The text always repeats "in our study", "our results", "According to our results", I believe that these expressions need to be improved.

Abstract

It needs to be improved globally. For example, information on the identified RM should be entered.

I would avoid starting the paragraph with "The originality".

Introduction

It needs to be improved and implemented. It must provide a review of the relevant literature that motivates the research question and a complete description of the experimental objectives and hypotheses.

Materials and Methods

2.1. Animals and Rearing Factors

Given the complexity, the paragraph needs improvement. Tables 1, 2 and 3 show the 30 factors, divided into the 3 breeding stages, which make it possible to identify the 5 RM; I believe that there is an overlap between "Materials and Methods" and "Results", in these tables the characteristics of each of the 30 identified factors should be specified, reporting the rest of the data in the "Results and Discussions" paragraph. This would allow each factor to be coded and to provide more information, improving readability.

Same indications for table 4. Furthermore, in "Results and Discussions" indicate why fewer animals were used for RM-3 and RM-4, 55 vs 58 and 31 vs 33 respectively.

Line 69, replace "CP" with "crude protein (CP)".

Line 199, replace "works" with "studies".

Lines 261-265, improve.

Line 262, "sudied", correct.

Conclusions, improve

Line 326, "as already observed for heifers", delete.

Best regards

Author Response

Reviewer 2

Open Review

English language and style

( ) Extensive editing of English language and style required
( ) Moderate English changes required
(x) English language and style are fine/minor spell check required
( ) I don't feel qualified to judge about the English language and style

Yes

Can be improved

Must be improved

Not applicable

Does the introduction provide sufficient background and include all relevant references?

( )

( )

(x)

( )

Are all the cited references relevant to the research?

( )

(x)

( )

( )

Is the research design appropriate?

( )

( )

(x)

( )

Are the methods adequately described?

( )

( )

(x)

( )

Are the results clearly presented?

( )

(x)

( )

( )

Are the conclusions supported by the results?

( )

(x)

( )

( )

Comments and Suggestions for Authors

Dear Authors,

the submitted manuscript evaluates the effect of different management factors at all stages of rearing on the final quality of the carcasses and meat. While acknowledging the originality of the study, the manuscript needs a thorough review before publication.

The text refers to the need "to produce simultaneously high carcass and LM meat qualities", this concept needs improvement;

To improve the reader’s understanding of the concept of joint enhancement of carcass and meat quality, we added elements at the end of the “Results and Discussion” section (L 373-379).

First of all, I believe that the aim is not to produce high quality LM meat, but to produce high quality meat. The investigations are carried out on LM because the characteristics of the muscle allow the tests to be repeatable and comparable.

We agree with this comment. We have modified the end of the introduction (L 48-50) and added element in the conclusion (L 391-393).

The text always repeats "in our study", "our results", "According to our results", I believe that these expressions need to be improved.

We have removed these expressions and rewritten the sentences to improve the text (L 18, 241, 247, 255, 257, 263, 264, 270-271, 318, 363)

Abstract

It needs to be improved globally. For example, information on the identified RM should be entered.

We have rewritten the abstract. In the new version of the abstract, the main characteristics of the both RM allowed the best trade-off to obtain high carcass and meat qualities simultaneously, were presented. (L 10-22)

I would avoid starting the paragraph with "The originality".

We have deleted “the originality” in the sentence (L 10)

Introduction

It needs to be improved and implemented. It must provide a review of the relevant literature that motivates the research question and a complete description of the experimental objectives and hypotheses.

We added element to improve and implement the introduction to better explain why we had performed this work. At the end of the introduction, we rewritten the aims and added hypotheses (L 32-52)

Materials and Methods

2.1. Animals and Rearing Factors

Given the complexity, the paragraph needs improvement. Tables 1, 2 and 3 show the 30 factors, divided into the 3 breeding stages, which make it possible to identify the 5 RM; I believe that there is an overlap between "Materials and Methods" and "Results", in these tables the characteristics of each of the 30 identified factors should be specified, reporting the rest of the data in the "Results and Discussions" paragraph. This would allow each factor to be coded and to provide more information, improving readability.

We have rewritten this paragraph (L 64-79) to explain how the rearing factors were calculated from the data of the rearing surveys. We added a new Table in the “Material and Methods” section to present the definition and the distribution of each rearing factors.

Same indications for table 4.

We added the description of the carcass traits in the “Materials and Methods” section (L 98-104) and in the table 5, we added the description of each carcass traits before ANOVA, as for the new Table 1. We performed also these modifications in the Table 7 to homogenize all result Tables presented in this article.

Furthermore, in "Results and Discussions" indicate why fewer animals were used for RM-3 and RM-4, 55 vs 58 and 31 vs 33 respectively.

First, we did not remove data, when the number were different the data were not available. There were many explanations for the variation in animal numbers for carcass and meat assessments. The missing data concerning the assessment on the carcass was explained by the fact that the assessments could not be performed by the slaughterhouse staff. The missing data concerning the meat analyses was explained by the fact that the analyses were performed during the COVID-19 period (cancelled sensory sessions while some samples were thawing) or the sample size did not allow to perform all meat analyses.

Line 69, replace "CP" with "crude protein (CP)".

We added this element in the text (L 73)

Line 199, replace "works" with "studies".

We have modified in the text (L 250)

Lines 261-265, improve.

Line 262, "sudied", correct.

We have rewritten this part to a better understanding of the reader (L 312-317)

Conclusions, improve

We have rewritten the conclusions of this study (L 381-393).

Line 326, "as already observed for heifers", delete.

We have deleted this port of the sentence (L 382)

Best regards

Round 2

Reviewer 2 Report

Dear Authors,

The manuscript has been improved and made easier to understand. In this form the manuscript, in my opinion, can be published in the journal.

Best Regards